# Automatic Labeling of Data for Transfer Learning

## Abstract

Transfer learning uses trained weights from a source model as the initial weights for the training of a target dataset. A well chosen source with a large number of labeled data leads to significant improvement in accuracy. We demonstrate a technique that automatically labels large unlabeled datasets so that they can train source models for transfer learning. We experimentally evaluate this method, using a baseline dataset of human-annotated ImageNet1K labels, against five variations of this technique. We show that the performance of these automatically trained models come within 17% of baseline on average.

## 1 Introduction

In many domains, the task performance of deep learning techniques is heavily dependent on the number of labeled examples, which are difficult and expensive to acquire. This demand for large labeled datasets has inspired alternative techniques, such as weak supervision or automated labeling, whose algorithms create plausible labels to be used to guide supervised training on other tasks.

In this work, we develop a content-aware model-selection technique for transfer learning. We take an unlabeled data point (here, an unlabeled image), and compute its distance to the average response of a number of specialized deep learning models, such as those trained for "animal", "person", or "sport". We then create a "pseudolabel" for the point, consisting of a short ordered sequence of the most appropriate model names, like "animal-plant-building". We use these synthetic labels to augment the ground truth labels. We validate the technique by applying it to the ImageNet1K dataset, as well as on a number of other large, unlabeled datasets.

## 2 Related Work

There are several well-established approaches that attempt to automatically assign labels to unlabeled images. For example, some use clusters of features to predict labels (Jeon et al., 2003), or augment image data with linguistic constraints from sources such as WordNet (Barnard et al., 2003; Jin et al., 2004). These approaches augment tasks by pretraining models using larger unlabeled data-sets. Pretraining approaches have also improved results when attempting a target task for which there is a limited amount of accurately labeled training data (Mahajan et al., 2018), by using *weakly* labeled data, such as social media hashtags, which are much more plentiful. However, effectiveness only appears to grow as the log of the image count. Further approaches use generative models such as GANs as in Radford et al. (2015) to explore and refine category boundaries between clusters of data, which exploit the rich statistical structures of both real and generated examples, sometimes augmented with labels or linguistic constraints. All of these automatic approaches use the structures present in large unlabeled datasets, to extend the expressivity of known labels, and to augment the raw size of training sets.

## 3 Approach

We present our technique using a specific case study involving images, and with source datasets created by vertically partitioning ImageNet22K (Deng et al., 2009) along its distinct subtrees: animal, plant, weapon, tools, music, fungus, sport, person, food, fruit, garment, building, nature, furniture,

vehicle, and fabric. We represent each such dataset by a single average feature vector. In this study, this vector is generated from the second last layer of a reference VGG16 (Simonyan & Zisserman, 2015) model trained on ImageNet1K, as shown in Figure 1, with the average taken over all the images in the dataset. To label a new image, we first calculate its own feature vector, then compute the distance between it and each of the representatives of the datasets; in this study, we use the Kullback-Leibler (KL) divergence, after appropriate normalization. These 16 distances are then used to determine the synthetic label.

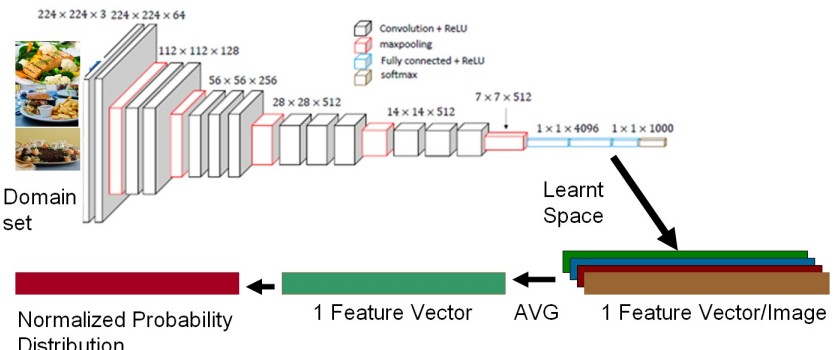

Figure 1: Feature Extraction using VGG 16 ImageNet1K trained model

# 4 LABELING METHODS

To label an image, we encode its relative positions with respect to some subset of the known labeled datasets. To better explain the technique, we will use ImageNet1K as a source of images and ground truth labels. ImageNet1K has 1000 labels; the number of images/label is almost uniform; but the labels broadly fall under few broad categories like animals, vehicles, food, musical instruments, garment, furniture, buildings etc. About 446 out of 1000 labels ($\sim$ 45%) belong to animals while the other top categories are vehicles (5.2%), food (2.2%), musical instruments (2.1%) and garment (2.1%). The distribution of number of labels for different categories has a long tail.

## 4.1 NEAREST-N LABELS

Our first labeling methods choose labels to be the names of the N source datasets which were the closest to the image. We concatenate those names together, in order of closeness. Thus, the Nearest-3 method generates labels like "tree-animal-fungus". With 16 source datasets, the Nearest-3 approach yields 16x15x14 = 3360 possible pseudolabels; in our study with the 1.3M images from ImageNet1K, each label had a mean of 381 images and a standard deviation of 1503. In a similar fashion, the Nearest-2 and Nearest-1 pseudolabels were also computed, with 240 and 16 possible labels, respectively. (We found no benefits above Nearest-3.) Figure 2a shows the distribution of images for the Nearest-3 method. The high peaks on the right are unsurprisingly related to animals.

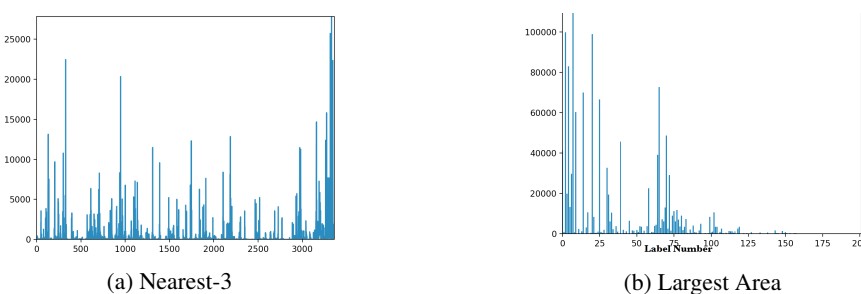

(a) Nearest-3          (b) Largest Area

Figure 2: Distribution of images per label in pseudo-labeled datasets

## 4.2 UNIFORM CLUSTERING

Our fourth method exploited all 16 distances. We first fixed the order of the 16 source datasets, forming a 16-dimensional coordinate system, so that each image could be viewed as a vector in the resulting 16-dimensional simplex of all possible positive distances to them. All of the unlabeled images were then k-means clustered within this space; we used 240 cluster centers to imitate the size of the Nearest-2 space. These resultant clusters could not be expected to be of uniform size, so a second round of splitting and merging, based on relative distances, balanced the cluster membership to be nearly uniform, with an average of 1000 images per cluster, and a standard deviation was 193. The "names" of these final clusters were used as the pseudolabels.

## 4.3 LARGEST AREA

Our fifth method accommodates those incoming datasets that are characterized by a wide variety of low-level image features, such as flowers. These labels were again devised as a sequence of three source names, but chosen to span the 16-dimensional space as widely as possible. The first source dataset was the closest one, and the second was the farthest one. The third was chosen from the remaining 14 to maximize the area of the resulting triangle in 16-space, computed via Heron's formula. In practice, this method only resulted in about 200 labels, with an average of about 6300 images per label, and with a very high standard deviation of about 17000 (see Figure 2b).

As an example, consider the images in Figure 3, taken from ImageNet1K. The nearest cluster (Nearest-1) for the body armour image is music. This is possibly due to the brass and metallic textures of the photo, similar to that seen in brass music instruments. The Nearest-2 and Nearest-3 labels are (music-weapon) and (music-weapon-person) respectively. In contrast, the label for Largest-Area is music-fungus-sport. So for this image, fungus is the source which is most unlike it, and sport maximizes the area of the triangle defined by the third source. Similarly, for the elephant image, the Nearest-3 label is tree-animal-fungus, and the Largest-Area label is tree-furniture-fungus. The Nearest-2 and Nearest-1 labels are tree-animal and tree, respectively.

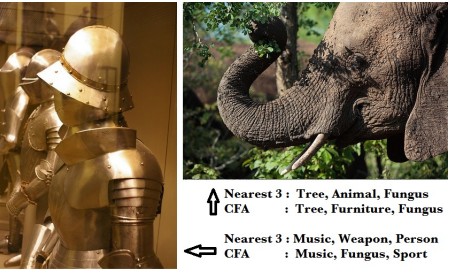

Figure 3: Pseudolabels assigned to example images from ImageNet1K

| dataset | labels | accuracy |
|---|---|---|
| Nearest-1 | 16 | 78.74% |
| Nearest-2 | 240 | 55.75% |
| Nearest-3 | 3360 | 37.07% |
| Uniform | 1144 | 33.49% |
| Largest-Area | 201 | 81.01% |
| Random | 1000 | 0.08% |
| Vanilla | 1000 | 67.14% |

Figure 4: Characteristics of different base model datasets, and their accuracy

## 5 EXPERIMENTAL EVALUATION

**Experiment 1** Using techniques described in Section 4, we first created five pseudo-labeled datasets for the images in ImageNet1K, as shown in Figure 4. We then trained ResNet27 using each of these pseudo-labeled datasets, creating base models for further transfer learning. We also created two baseline models, one using the vanilla ImageNet1K dataset of images and human-annotated labels, and a second by assigning random labels to the ImageNet1K images. For perspective, the figure shows the accuracy of these seven base models, but since we are interested in the *transferability* of representations from these base models to a target domain, their absolute accuracy is not the important measure. Then, for each of 12 candidate target datasets, we fine-tuned and calculated the transfer learning accuracy of each of the 7 base models; each of these 84 experiments were carried out with the same hyperparameters.

As shown in Figure 4, the accuracy obtained with *Vanilla* as the base model can serve as an upper-bound on transfer learning accuracy for experiments using *Nearest-N, Uniform,* and *Largest-Area* base models. Similarly, *Random* can provide a lower bound. Table 1 shows that *Nearest-3* gives

the best accuracy for 8 out of 12 datasets. For two datasets, Nearest-2 performs slightly better than Nearest-3, while Uniform and Largest-Area perform best for the person and the flowers datasets.

**Experiment 2**  To capture the performance of pseudo-labeled vs. human-labeled datasets, we define in the usual way the relative error of transfer learning accuracy between a pseudo-labeled dataset, $i$, and the *Vanilla* base model, $v$, as: $Err_i = (1 - accuracy_i/accuracy_v) \times 100\%$. For each target dataset, we also calculate their KL divergence with respect to ImageNet1K, as defined in Section 3.

| Base → | Pseudo-labeled | | | | | | Imagenet1K |
|---|---|---|---|---|---|---|---|
| Target ↓ | Nearest-1 | Nearest-2 | Nearest-3 | Largest-Area | Uniform | Random | Vanilla |
| music | 42.98% | 43.60 % | **43.86%** | 42.87 % | 43.71% | 1.57% | 47.19% |
| tool | 38.79% | 39.12% | **39.44%** | 39.40% | 39.39% | 1.24% | 42.65% |
| weapon | 29.51% | **30.24%** | 30.21% | 29.46% | 29.92% | 2.09% | 32.25% |
| fungus | 21.28% | 21.96% | **22.16%** | 21.78% | 21.88% | 1.60% | 23.59% |
| flowers | 75.94% | 74.90% | 72.88% | **76.64%** | 72.36% | 0.43% | 85.13% |
| sport | 28.68% | 30.46% | **30.76%** | 30.01% | 30.74% | 0.98% | 37.37% |
| person | 6.87% | 7.25% | 7.89% | 7.29% | **8.05%** | 0.12% | 10.12% |
| food | 8.58% | 9.21% | **9.62%** | 9.19% | 9.36% | 0.13% | 12.52% |
| fruit | 18.53% | **19.53%** | 19.12% | 19.05% | 18.54% | 0.82% | 25.95% |
| garment | 16.84% | 17.30% | **18.05%** | 17.29% | 17.61% | 0.68% | 24.48% |
| animal | 15.40% | 17.54% | **18.46%** | 17.31% | 18.40% | 0.10% | 24.87% |
| plant | 10.46% | 11.27% | **11.69%** | 11.14% | 11.47% | 0.08% | 15.34% |

Table 1: Transfer learning accuracy for different target datasets; best accuracy shown in bold.

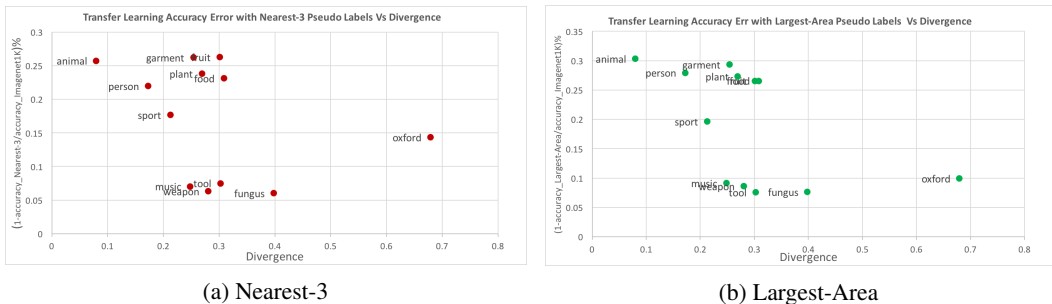

(a) Nearest-3         (b) Largest-Area

Figure 5: Relative error between pseudo-labels and Vanilla, vs. divergence, for 12 target datasets

Figures 5a and 5b show plots of $Err_i$ for 12 different target datasets. The average value of $Err_i$ is 17.2%, with minimum and maximum being 6.1% and 26.3%. Thus, using base models trained with automatically generated labels, transfer learning accuracy is on average only 17.2% worse when compared to base models trained using human labeled images. Further, the error shrinks when the divergence increases. This implies that when the base dataset is far away in feature space, the transferability of representations is less sensitive to noise in the labels.

## 6    CONCLUSION

We have shown that generation of content-aware pseudolabels can provide transfer performance approaching that of human labels, and that models trained on psuedolabels can be used as source models for transfer learning. The automated approach presented here suggests that the internal representations of content models trained on specialized datasets contain some descriptive features of those datasets. By treating each of these specialized representations as a "word" in a longer "sentence" that describes a category of images, we can create labels such as a "music-weapon-person" to describe a suit of armor, or a "tree-animal-fungus" to describe an elephant. These rich labels capture features of these objects such as visual information about the materials they are made out of, that better describe the contents than reliance on a single label would produce. Using multiple, content-aware models to achieve greater descriptive power may be a valuable future avenue of research.

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
