# OpenReview forum: "Automatic Labeling of Data for Transfer Learning"
_ICLR.cc/2019/Workshop/LLD — Submitted to LLD 2019_

### Official Review · AnonReviewer2 · 2019-04-12
**Variations on pseudo-label generation**

**Rating:** 2
**Confidence:** 2

**Review:**

This paper describes 5 methods for generating pseudo-labels for images using a pre-trained VGG net. They involve concatenating N-nearest labels, clustering, and a geometric method.

- The first three methods might also have been classified as three variants of N nearest neighbors with N=1,2,3

- The geometric method creating a maximum surface triangle doesn't appear to have any motivation

- An important contribution in this domain, Hsu et al, 2018, https://arxiv.org/abs/1810.02334, which performs clustering on neural network features, bearing a fair amount of similarity to at least method 4, is not mentioned in this paper

- The specific notion of "KL divergence" is not explained or stated as a formula, making the exact probability distributions over which it is computed impossible to know

- Minor: "figure 2" should be a table (but appears to be a cropped screenshot from a spreadsheet)

While it is good that different types of pseudolabeling are explored and evaluated, this paper is lacking on many fronts.

---

### Official Review · AnonReviewer1 · 2019-04-14

**Rating:** 2
**Confidence:** 2

**Review:**

The main idea of the paper is to algorithmically label (or pseudolabel) a large amount of image data so that the model pre-trained on these data can better transfer to some target task. The idea is to use some specialized (also pre-trained) models that can identify certain classes as soft labeling functions (i.e., at pre-training time, use the distance to the average output activations as the pre-training learning signal).

The idea is intuitive and seems working, but the downside of the approach is that it requires these gold-standard specialized models used for source labeling (or known labeled datasets used to construct such models). Of course, this is just a pre-training method, and the goal is to transfer the model to a target domain that has some (limited) gold-standard annotations. However, the approach seems quite heuristic, it's unclear whether such pre-training introduces any biases (during pre-training), and generally a more thorough analysis of how the quality of the known labeled data affects model pretraining is necessary.

---

### Decision · Program_Chairs · 2019-04-15
**Acceptance Decision**

Reject